# Combination of a Novel Fusion Protein CD3εζ28 and Bispecific T Cell Engager Enhances the Persistance and Anti-Cancer Effects of T Cells

**DOI:** 10.3390/cancers14194947

**Published:** 2022-10-09

**Authors:** Feng Yu, Yang Gao, Yan Wu, Anran Dai, Xiaoyan Wang, Xiangzhi Zhang, Guodong Liu, Qinggang Xu, Dongfeng Chen

**Affiliations:** 1School of Life Science, Jiangsu University, No. 301 Xuefu Road, Zhenjiang 212013, China; 2Department of Gastroenterology, The Affiliated Suqian First People’s Hospital of Nanjing Medical University, Suqian 223812, China

**Keywords:** BiTE, CD3, CD28, costimulatory, immunotherapy

## Abstract

**Simple Summary:**

Bi-specific T cell engager (BiTE) has shown promising therapeutic potential in preclinical and clinical studies. However, T cells cannot be sufficiently activated by BiTE, most likely due to lacking co-stimulatory signal. We, therefore, designed a chimeric fusion protein, named as CD3εζ28, which consists of the CD3ε extracellular region, the CD28 costimulatory signal and the intracellular region of CD3ζ in tandem. Our results demonstrated that T cells genetically modified to express CD3εζ28 could be sufficiently activated and effectively kill tumor cells in presence of BiTE molecules in vitro, and, thus, show superior anti-tumor effects on xenograft tumor models. Taken together, incorporating co-stimulatory signal may be an effective approach to improve the effector function of T cells mediated by BiTE.

**Abstract:**

Bi-specific T cell engager (BiTE), an artificial bi-functional fusion protein, has shown promising therapeutic potential in preclinical and clinical studies. However, T cells cannot be sufficiently activated by BiTE, most likely due to lacking co-stimulatory signal. We reasoned that incorporating co-stimulatory signal might have the potential to enhance the T cell activation mediated by BiTE. We, therefore, designed a chimeric fusion protein, named as CD3εζ28, which consists of the CD3ε extracellular region, the CD28 costimulatory signal and the intracellular region of CD3ζ in tandem. T cells genetically modified to express both *CD3εζ28* and *GFP* (T-CD3εζ28-GFP) were generated by retroviral transduction. The results from in vitro experiments showed that T-CD3εζCD28-GFP cells had superior cytotoxic effects on tumor cells in presence of BiTE compared with control T cells, as evidenced by IL-2 and IFN-γ production, T cell proliferation and sequential killing assay. In vivo, T-CD3εζCD28-GFP cells showed superior anti-tumor effects in Hela-BiTE. EGFRvIII xenograft tumor model, as evaluated by tumor growth rate and T cell persistence in comparison with control T cells. In order to further confirm these findings, we generated T cells modified to express both *CD3εζCD28* on cell surface and BiTE.CD19 by autocrine manner (T-CD3εζCD28-BiTE.19). The superior anti-tumor effects of T-CD3εζCD28-BiTE.19 cells could also be evidenced by the similar in vitro and in vivo experiments; thus, incorporating co-stimulatory signal may be an effective approach to improve the effector function of T cells mediated by BiTE.

## 1. Introduction

T cells are the main effector cells for cell-mediated immune response targeting tumor cells, so the optimal modifications of T cells have become the main object of most targeted cancer therapies [1]. The activation of T lymphocytes requires at least two signals. Signal one is delivered by the T-cell receptor (TCR)-CD3 complex whereas signal two is provided by co-stimulatory molecules, such as 4-1BB or CD28 [2,3]. In the presence of signal one, but absence of costimulatory signals, T cells cannot be completely activated, which leads to T cell anergy or apoptosis [4]. Among the different determined costimulatory pathways, the most critical stimulatory signals are mediated by CD28 on T cell surface, which can interact with its ligands CD80 or CD86 from antigen presenting cells (APCs) [5].

Bi-specific T cell engagers (BiTEs), a class of artificial bi-specific monoclonal antibodies, have shown remarkable therapeutic effects for hematological malignancies and some certain solid tumors [6,7]. BiTE molecules consist of two distinct single-chain variable fragments (scFvs) that can recognize antigens on tumor cells and CD3ε on T cells, respectively. The intracellular signaling domain of CD3ε contains an immunoreceptor tyrosine-based activation motif (ITAM) domain, which can activate and evoke T cells to lyse tumor cells in the presence of BiTEs [8,9,10]. In principle, T cells evoked by BiTEs can persistently kill target tumor cells. However, due to lacking co-stimulatory signaling, T cells cannot be fully activated by BiTEs, which may lead to T cells anergy [9,10]. Involvement of co-stimulatory signal has been proved as a promising solution to enhance T cell activation, and therefore improving BiTE efficacy in cancer therapy. Colin E. Correnti, et al. generated a modified BiTE that recognized CD28, named as CD28-BiTE [10]. Combining it with the CD3-BiTE could significantly enhance T cell activation, Mireya Paulina Velasquez, et al. found that T cells expressing both BiTE and co-stimulatory signals 41BBL/CD80 could be completely activated and showed superior anti-tumor effects [11].

In this study, we deigned a new fusion protein that composes of the CD3ε extracellular region, the CD28 costimulatory signal and the intracellular region of CD3ζ in tandem, named as *CD3εζ28*. Furthermore, we explored if combination of CD3εζ28 and BiTE could sufficiently activate T cells and enhance their anti-tumor effects both in in vitro and in vivo.

## 2. Materials and Methods

### 2.1. Cell Lines

The cells lines Hela, A549 and 293T were cultured in high-glucose DMEM (ThermoFisher Scientific, Waltham, MA, USA) supplemented with 10% fetal bovine serum (FBS) (Shanghai life iLAB BIO Technology, Shanghai, China). The cells lines Nalm6, Raji, Daudi, K562, were maintained in conditioned RPMI 1640 (Thermo Fisher, Waltham, MA, USA) containing 10% FBS. The stable cells lines, Hela-EGFRvIII-BiTE, Nalm6-ffluc-GFP, Raji-ffluc-GFP, Daudi-ffluc-GFP, K562-ffluc-GFP and K562-CD19-ffluc-GFP were established by lentiviral transduction and purchased from Vigen Biotechnology (Vigen Biotechnology, Zhenjiang, Jiangsu, China). The contamination of mycoplasma were routinely detected in all cell lines.

### 2.2. BiTE Generation

The EGFRvIII-BiTE plasmid contains the DNA sequences expressing immunoglobulin heavy-chain leader peptide, the EGFRvIII-specific scFv 806, (G4S)3 linker and a CD3-specific scFv. The CD19-BiTE plasmid was constructed by replacing EGFRvIII-specific scFv 806 with CD19 specific scFv FMC63. For generation of BiTE supernatant, the recombinant plasmids carrying BiTE were transfected into 293T cells, and the culture supernatants of 293T cells were collected 48 h later.

### 2.3. Construction of Retroviral Vectors

The CD3εζ28 fusion gene that consists of CD3ε extracellular domain, SH (short Hing of IgG1), CD28 intracellular domain, CD3ζ intracellular domain was synthized and subcloned into pSFG-MCS-T2A-GFP or pSFG-MCS-T2A-mOrange backbone. The BiTE.CD19 was subcloned into pSFG-MCS-IRES-mOrange or pSFG-MCS-E2A-CD3εζ28-T2A-mOrange backbone, separately. RD114-pseudotyped retroviral particles were generated by co-transfection of shutter plasmids with packaging plasmid, pRD114 and Peq-pam 3 (-E).

### 2.4. T Cell Stimulation 

The blood samples from healthy human donors were collected according to the protocol approved by the Ethics Committee of Jiangsu University. All the donors signed consent. The peripheral blood mononuclear cells (PBMCs) were isolated using lymphocyte isolation kit (TBD Bio, Tianjin, China). Then PBMCs were cultured in a T551 medium (Takara, Japan) containing 10% FBS (Gibco) with 300 IU/mL IL-2 (Intekang, Suzhou, Jiangsu, China). PBMCs were stimulated by anti-CD3 (C587, Novoprotein, Suzhou, Jiangsu, China) (1 mg/mL) and anti-CD28 (CI67, Novoprotein, Suzhou, Jiangsu, China) (1 mg/mL) with recombinant human IL-2 for two days to activate T cells. 

### 2.5. Generation of Modified T Cells

Activated T cells were mixed with retrovirus carrying GFP, CD3εζ28-GFP/mOrange (mO), BiTE.19-mO or CD3εζ28-BiTE.19-mO. For each tube, 50 μL concentrate of retrovirus were added into 2.5 × 10^5^ cells in 100 μL T551 medium (Takara, Japan). The mixtures were centrifuged at 400 g/min for 50 min. Next, we directly added 350 μL T551 medium containing 10% FBS into each tube, while the cell suspension was seeded into retronectin-coated 24 wells plate and cultured in 37 °C 5% CO_2_ for further experiments.

### 2.6. Flow Cytometry

Cells were washed once with PBS and incubated with indicated antibodies for 30 min on ice in the dark. After washing once with PBS again, 10,000 cells per sample were analyzed using a FACS Calibur instrument (Beckman CytoFLEX S, Brea, CA, USA). The GFP and mOrange fluorescence signal was detected using FITC and PE-A channel, respectively. APC-anti-CD3 (340440, BD Bioscience, Franklin Lakes, NJ, USA) was used to detect the T cells in tumors. APC-anti-CD19 (HIB19, Biolegend, San Diego, CA, USA) was used to detect the CD19 positive cancer cells. The 7AAD dye (BD Biosciences, Franklin Lakes, NJ, USA) was used to detect apoptosis in T cells.

### 2.7. Enzyme-Linked Immunosorbent Assay (ELISA)

T cells were co-cultured with tumor cells at different time points. Then the supernatants were collected and analyzed by human IL-2/IFN-γ quantity ELISA kit (BD Bioscience, Franklin Lakes, NJ, USA) according to the manufacturer’s instructions. Data were obtained by detecting the absorbance value of the sample at the indicated wavelength. 

### 2.8. Cytotoxicity Assay

Adherent tumor cells were co-cultured with T cells in 96-well plates for 24 h. After removing the supernatant, the plates were gently washed twice with PBS. The viability of tumor cells was then determined using an MTS formazan viability assay (Promega, Fitchburg, WI, USA). Briefly, 5 μL MTS solution and 95 μL medium were added into each well and incubated at 37 °C for one hour. Optical density of each well was determined at 490 nm on a microplate reader (Molecular Devices VERSAmax, Sunnyvale, CA, USA). Suspended cancer cells stably expressing the ffluc were co-cultured with T cells in 96-well plates for 24 h and then added 0.5 μL luminescent substrate (Yishen Technology, Shanghai, China) and 50 μL medium. The fluorescence intensity of each well was determined using luminometer Lumistation1800 (Shanpu Biotechnology, Shanghai, China).

### 2.9. Xenograft Model

Four-week-old female nude mice (Carvens, Changzhou, Jiangsu, China) were injected subcutaneously with 5 × 10^6^ tumor cells per mouse. After 3 days, T cells were injected intratumorally with 1 × 10^7^ cells per mouse. Then tumor sizes were measured with calipers every two days, and tumor volumes (in mm^3^) were determined using the formula W^2^ × L/2, where W and L represents tumor width and tumor length, respectively.

### 2.10. Immunohistochemistry (IHC) Staining

Tumors were paraformaldehyde fixed, embedded in paraffins, and sectioned into 5 µm sections. The sections were then stained with anti-CD3 (C587, Novoprotein, Suzhou, Jiangsu, China) according to the manufacture’s guidelines. Images were captured and analyzed using microscope (OLYMPUS, Tokyo, Japan) and Images Pro Plus 6.0. software, respectively.

### 2.11. Statistical Analysis

Data were analyzed using Prism 8.0 (GraphPad, San Diego, CA, USA) software and the results were presented as mean ± SE where indicated. Statistical significance was determined by using multiple comparison *t*-test. *p* < 0.05 is considered as significant difference (* *p* < 0.05, ** *p* < 0.01, *** *p* < 0.001).

## 3. Results

### 3.1. Generation of T Cells Expressing CD3εζ28

To obtain T cells expressing *CD3εζ28*, the fusion gene was cloned into pSFG retroviral backbone containing a GFP gene (Figure 1A). T cells were transduced with the above retrovirus named as T-CD3εζ28-GFP and T-GFP, respectively. The transduction rates were detected both by flow cytometry (FCM) and fluorescence microscopy after five days. As shown in Figure 1B, the transduction rates of both T-CD3εζ28-GFP and T-GFP were above 60%, which was consistent with the results observed under fluorescence microscopy (Figure 1C). To check the expression of CD3εζ28, quantitative real-time PCR (qRT-PCR) and western blotting (WB) were performed. The qRT-PCR results showed that the relative mRNA levels of CD3εζ28 were significantly higher in T-CD3εζ28-GFP cells than those in non-transduced activated T cells (nATC) and T-GFP cells (Figure 1D). The WB results revealed that the endogenous CD3ζ chain was found in all types of indicated T cell while CD3εζ28 was only found in T-CD3εζ28-GFP cells (Figure 1E). To determine the effect of CD3εζ28 on T cell proliferation, the cell number was counted at indicated time. The results showed that the number of T-CD3εζ28-GFP cells were significant higher than other groups at day 14 and day 21 post-transduction (Figure 1F), which indicates that CD3εζ28 can enhance T cell proliferation.

### 3.2. Combining T-CD3εζ28-GFP with BiTEs Exhibited Synergistic Killing Effects on Tumor Cells

Previous study demonstrated that the EGFRvIII is highly expressed on the cell surface in HeLa cell line [12], which indicates that it is an ideal immunotherapy target. Choi et al. designed a BiTE that can specifically recognize EGFRvIII, named as EGFRvIII-BiTE [13]. Here we generated the EGFRvIII-BiTE in 293T cells by transient transfection. To assess the anti-tumor efficacy of EGFRvIII-BiTE, HeLa cells were co-cultured with the different effector T cells with or without EGFRvIII-BiTE. The results from MTS assay showed that T-CD3εζ28-GFP displayed the strongest cytotoxicity to tumor cells when combined with EGFRvIII-BiTE comparing with the control T cells (Figure 2A,B). To further confirm the findings, the HeLa cell line stably secreting EGFRvIII-BiTE (HeLa-EGFRvIII-BiTE) was established by lentiviral transduction (Appendix A) and co-cultured with T-CD3εζ28-GFP, T-GFP or nATC with indicated ratios. As expected, T-CD3εζ28-GFP showed strongest cytotoxicity to Hela-EGFRvIII BiTE tumor cells according to the results from MTS assay (Figure 2C). To further test whether combination of T-CD3εζ28-GFP with other BiTEs has a similar anti-tumor effect, an A549 cell line stably secreting EphA2-BiTE was generated by lentiviral transduction and named as A549-EphA2-BiTE (Appendix A). MTS assay revealed that T-CD3εζ28-GFP had better killing effects on A549-EphA2-BiTE comparing with the control T cells (Appendix A). To investigate whether T-CD3εζ28-GFP cells were activated by tumor cells secreting BiTEs, the concentrations of IL-2 and IFN-γ in the supernatant from the co-culturing medium of effector T and HeLa-EGFRvIII-BiTE cells were detected using enzyme-linked immunosorbent assay (ELISA). The results showed that the concentrations of IL-2 and IFN-γ secreted by T-CD3εζ28-GFP were significantly higher (*p* < 0.05) than them in the control groups at indicated ratios (Figure 2D,E), indicating that T-CD3εζ28-GFP has higher levels activation. The same findings were also found when T-CD3εζ28-GFP cells were co-cultured with A549-EphA2-BiTE cells (Appendix A). To further confirm these findings, the relative mRNA expression levels of T cell activation-related genes were examined using qRT-PCR. Compared with nATC and T-GFP, the mRNA levels of CD69, Granzyme B, CD107α, IFN-γ and TNF-α were significantly increased in T-CD3εζ28-GFP cells (*p* < 0.05) when co-cultured with HeLa-EGFRvIII-BiTE (Figure 2F). Taken together, these results manifested that T-CD3εζ28-GFP could be better activated by BiTEs, therefore, showing the higher cytotoxicity to tumor cells. Since T-CD3εζ28-GFP cells showed stronger activation and cytotoxicity, we predicted that activated T-CD3εζ28-GFP cells might persist longer than nATC and T-GFP cells when repeated exposure to target tumor cells. To confirm this, different effector T cells were repeatedly stimulated by HeLa-EGFRvIII BiTE with same E:T ratio every 7 days, and T cell proliferation was evaluated by cell number counting. The results showed that T-CD3εζ28-GFP revealed superior proliferation and persistence following repeated stimulating with target tumor cells (Figure 2G,H).

### 3.3. T-CD3εζ28-GFP Cells Showed Enhanced Anti-Tumor Activity in HeLa-EGFRvIII-BiTE Xenograft Tumor Model

To investigate the anti-tumor effects of T-CD3εζ28-GFP in vivo, a xenograft tumor model was established by subcutaneously injecting HeLa-EGFRvIII-BiTE cells into the right flank of nude mice. Effector T cells were intratumorally injected at the third day after inoculation of tumor cells (Figure 3A). Then the sizes of tumors were measured every two days. The growth rate of tumors in T-CD3εζ28-GFP treated group was significantly slower than those in PBS, nATC and T-GFP groups (*p* < 0.05) (Figure 3B). Meanwhile, the sizes and weights of tumors from the T-CD3εζ28-GFP treated group at the endpoint of experiment were significantly less than them from control groups (*p* < 0.05) (Figure 3C,D). Moreover, the total weight of the mice was not affected by T-CD3εζ28-GFP cells treatment (Figure 3E). These results indicate that T-CD3εζ28-GFP cells had super anti-tumor efficacy and without obvious side effects in vivo.

To evaluate the persistence of different types of indicated T cell in vivo, the total cells from tumor tissues were collected after 9 days of T cells injection and stained with anti-CD3 antibody to determine the percentages of human T cells in dissected tumor tissues by FCM. The results showed that the percentage of T-CD3εζ28-GFP cells in tumors could achieve 12% and much higher than those in nATC and T-GFP treated groups (Figure 3F). Moreover, the results from immunohistochemistry (IHC) staining also showed much more CD3 positive human T cells in T-CD3εζ28-GFP treated tumor than those in nATC and T-GFP treated groups (Figure 3G), and the mean integral optical density (IOD) in the T-CD3εζ28-GFP group was significantly higher than it in the control group (Figure 3H). Taken together, our findings demonstrated that T-CD3εζ28-GFP could long-time persist in vivo.

### 3.4. Generation of T-CD3εζ28 Cells Expressing Autocrine BiTEs

For clinical applications, separately using T-CD3εζ28 cells and BiTE proteins increase the complications for treatment, which may lead to difficulties to make appropriate treatment regime. Therefore, to overcome these issues, we generated T-CD3εζ28 cells expressing autocrine CD19 BiTE (T-CD3εζ28-BiTE.19) and T cells expressing CD19 BiTE (T-BiTE.19) by retroviral transduction (Figure 4A). After 5 days post-transduction mOrange expression was detected by FCM analysis. Over 80% of cells were mOrange positive (Figure 4B). The results from the western blotting showed that the expression of the fusion protein CD3εζ28 could be found only in T-CD3εζ28 and T-CD3εζ28-BiTE.19 (Figure 4C). In addition, the proliferation assay showed that the T-CD3εζ28 and T-CD3εζ28-BiTE.19 cells grew significantly faster than other types of T cells (Figure 4D). To verify whether the secreted CD19-BiTE could mediate T cell killing, the cell supernatants were collected from each group. The nATC cells or the CD19 over-expressing Hela cells (Hela-CD19) were co-cultured in the presence of different supernatant. The results showed that only the supernatants from T-BiTE.19 or T-CD3εζ28-BiTE.19 cells could effectively induce T cell mediated killing effects on Hela-CD19 cells (Figure 4E).

### 3.5. T-CD3εζ28-BiTE.19 Cells Exhibited Superior Killing Effects on Cancer Cells

To further determine the killing effects of T-CD3εζ28-BiTE.19 on cancer cells, Nalm6, Raji, Daudi and K562-CD19 (Appendix A) were co-cultured with different types of T cells at indicated E:T ratios. The results showed that T-CD3εζ28-BiTE.19 and T-BiTE.19 cells could specifically kill cancer cells, whereas nATC and T-CD3εζ28 cells had almost no killing effects due to lacking CD19-BiTE, and that T-CD3εζ28-BiTE.19 had a stronger cytotoxicity than T-BiTE.19 cells (Figure 5A). To investigate if T-CD3εζ28-BiTE.19 cells are highly activated when co-cultured with cancer cells, the concentrations of IFN-γ and IL-2 in cell culture supernatants were determined by ELISA in Nalm-6 cell group. The results showed that the T-CD3εζ28-BiTE.19 cells secreted highest levels of IFN-γ and IL-2 among the different T cell groups at indicated E:T ratios (Figure 5B). To determine the persistence of the T-CD3εζ28-BiTE.19 cells, different effector T cells were repeatedly stimulated by HeLa-CD19 with E:T ratio 5:1 every 3 days for 3 rounds. The T cell proliferation and apoptosis were evaluated by and the T cell numbers in each round were counted using a Burker chamber hemocytometer. The results showed that T-CD3εζ28-BiTE.19 cells revealed strongest capacity of proliferation and anti-apoptosis comparing with other types of T cell following repeatedly stimulating with HeLa-CD19 cells (Figure 5C,D). These findings suggest that T-CD3εζ28-BiTE.19 cell have superior capacity of proliferation and persistence in vitro.

### 3.6. T-CD3εζ28-BiTE.19 Cells Showed Enhanced Anti-Tumor Activity in Xenograft Tumor Model

To verify the anti-tumor effect of T-CD3εζ28-BiTE.19 in vivo, a CD19 positive xenograft tumor model was established by subcutaneously inoculation of HeLa-CD19 cells into the right flank of nude mice, and T cells were intratumorally injected after three days. The sizes of tumors were measured every two days. The growth rate of tumors in T-CD3εζ28-BiTE.19 and T-BiTE.19 treated group were significantly slower than nATC and T-CD3εζ28 groups (*p* < 0.0001) (Figure 6A). Meanwhile, the sizes and weights of tumors from the T-CD3εζ28-BiTE.19 and T-BiTE.19 treated group at the experiment endpoint were significantly less than those from control groups (*p* < 0.001), and the sizes and weights of tumors from the T-CD3εζ28-BiTE.19 group were the least among all groups (*p* < 0.05) (Figure 6B,C). Moreover, the total weight of the mice was not affected by any indicated treatments (Figure 6D). These results suggest that T-CD3εζ28-BiTE.19 cells can effectively inhibit tumor growth in vivo without causing obvious side effects.

## 4. Discussion

Activation and persistence have become the key factors affecting the functions of T cells for cancer immunotherapy. It has been reported that the BiTE can mediate T cell activation to specifically kill cancer cells. Hence, BiTE has become one of the most potential strategies for cancer immunotherapy [6]. However, due to lacking co-stimulatory signaling, T cells cannot be completely activated by BiTE [7]. Here we designed a novel fusion protein CD3εζ28 to enhance the BiTE-mediated T cell activation. Our results showed that when combined with BiTE, T cells carrying CD3εζ28 had better killing effects on tumor cells and expressed higher levels of IL-2 and IFN-γ and the genes related to T cell activation than normal T cells. In addition, after activated by BiTE, T cells carrying CD3εζ28 could persistently survive and exert killing effects on tumor cells both in vitro and in vivo.

T cell receptor (TCR)-CD3 complex includes antigen recognition subunit αβ, and three signaling subunits CD3εδ, εγ and ζζ. Intracellularly, all three signaling subunits contain immune receptor tyrosine motifs (ITAMs) that initiate the downstream signaling via tyrosine phosphorylation [3]. Generally, BiTE binds to CD3ε subunits, and then leads to the recruitment of ZAP70 and the activation of T cells after CD3ε ITAM phosphorylation by tyrosine kinase (LCK) [2,3,6]. Notably, the intracellular region of the CD3ε subunit has only one ITAM while the intracellular segment of the CD3ζ subunit has three ITAMs [3], which means that CD3ζ can transmit more T cell activation signals to downstream. However, Thistlethwaite et al. found that only CD3ζ could not enhance the activation of T cells and not be enough to inhibit the apoptosis of T cells, which means that the extra stimulatory signals may be needed to resolve these issues [14]. Our novel fusion protein CD3εζ28 cannot only introduce co-stimulatory signal into the activation pathway but also use CD3ζ to transmit more activation signals.

The intracellular region of CD3εζ28 consists of the CD3εζ subunit and the CD28 costimulatory signal, which is similar to the intracellular region of chimeric antigen receptor (CAR) [6,15]. Meanwhile, combining the extracellular domain of CD3ε and BiTE conferred a tumor-specific role on T cells, which function similar to CAR-T cells. However, the tumor cell specific recognition function and activation of CAR-T cells are strictly dependent on CAR, so the transduction efficiency of CAR largely limits the killing effects of CAR-T cells on tumor cells, and the un-transduced T cells almost have no killing effects on tumor cells. Compared to CAR-T cell therapy, the secretory property of BiTE can bring bystander effect, which endows the surrounding T cells with the function of specific recognition and killing tumor cells [6,15]. In this study, T cells carrying CD3εζ28 were used in combination with BiTE to compensate for the deficiency of BiTE-induced T cell activation, thereby enhancing the therapeutic effect of BiTE. Last but not least, this T-cell engineering strategy may be further improved by linking CD3εζ28 with genes encoding cytokines chemokines, prodrug-converting enzymes or antiangiogenic agents, all of which have been reported to improve CAR-T cell therapy and showed enhanced anti-cancer effects [16,17,18,19,20,21].

## 5. Conclusions

In conclusion, our findings demonstrated that the engineered T cells expressing CD3εζ28 and/or BiTE could be highly activated and revealed strong antitumor activity in xenograft tumor models, which provides preclinical evidence for the therapeutic potential of CD3εζ28 fusion protein.

## Figures and Tables

**Figure 1 cancers-14-04947-f001:**
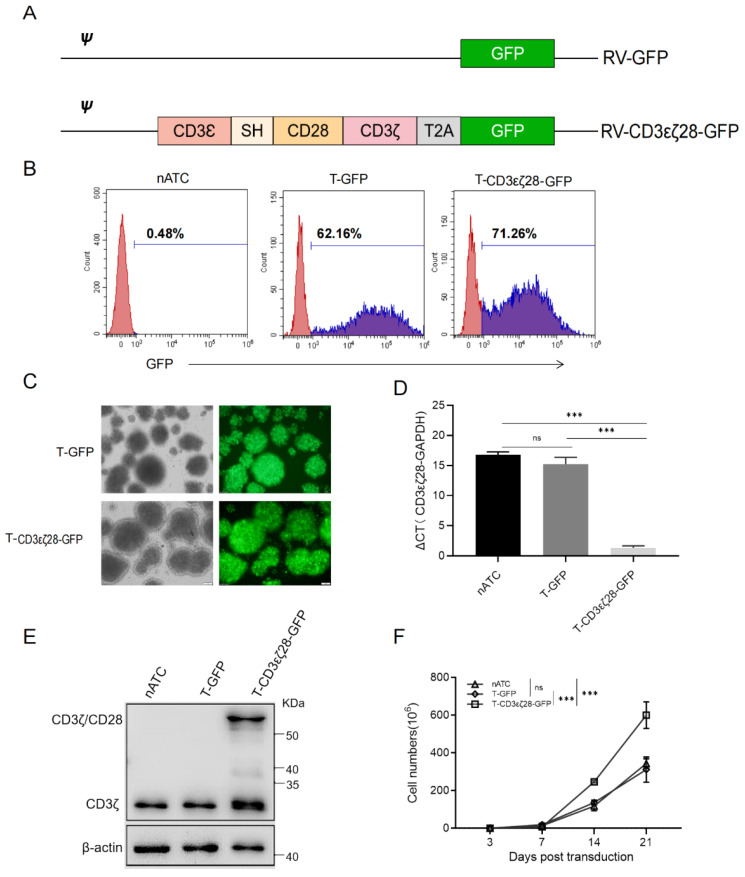
Generation of T-CD3εζ28-GFP cells. (**A**) The expression cassettes of T-CD3εζ28-GFP cells and T-GFP. (**B**) Flow cytometry analysis showed the transduced rate of retrovirus carrying CD3εζ28-GFP or GFP in T cells based on the expression of GFP. In this case, nATC was a non-transduced control. (**C**) Transduced T cells were observed by fluorescence microscopy on the fifth day post transduction. (**D**) RT-qPCR showed the relative expression levels of the fusion gene CD3εζ28.in the indicated T cells. (**E**)The expression levels of endogenous CD3ζ and CD3εζ28 fusion protein in different T cells were evaluated by western blotting with anti-CD3ζantibody. (**F**) The growth curves of different T cells were made based on the cell number counting at indicated timepoint for 21 days in vitro. Statistical differences were assessed by mutual comparison between each two groups using multiple comparisons *t* test. (*** *p* < 0.001, ns: no significance).

**Figure 2 cancers-14-04947-f002:**
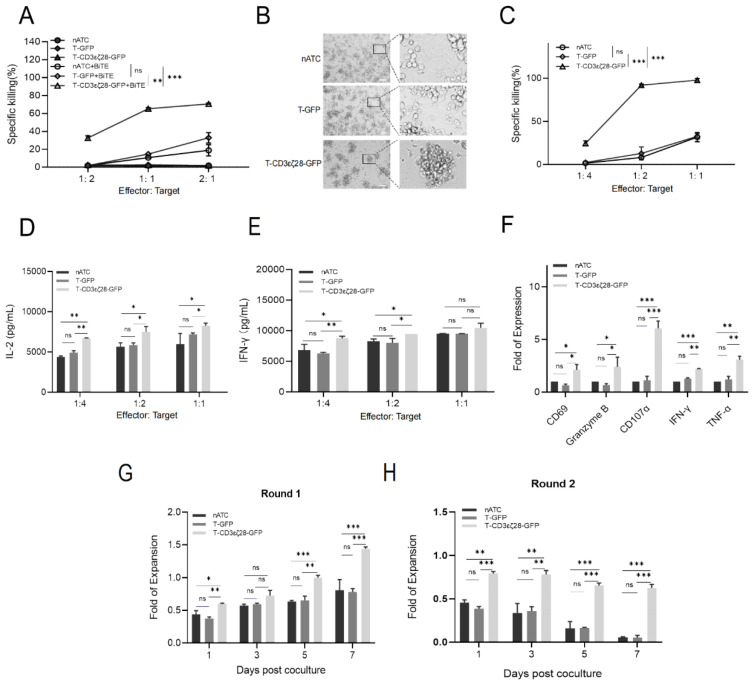
Combination of T-CD3εζ28-GFP cells and BiTE exhibited superior capacities of killing cancer cells and persistent proliferation. (**A**,**B**) Comparison of the cancer cell killing effects of different T cells with or without added BiTE. HeLa cells were co-cultured with nATC or T-GFPor T-CD3εζ28-GFP at indicated effector to target (E:T) ratios, with or without EGFRvIII-BiTE for 24 h. The viability of cancer cells was determined by using MTS assays (**A**). The cancer cell killing effects of different T cells with EGFRvIII-BiTE were observed under microscope at E:T ratio of 1:1 (**B**). (**C**) Comparison of the killing effects of different T cells on cancer cells expressing BiTE. Hela cells expressing EGFRvIII BiTE (HeLa-EGFRvIII BiTE) were co-cultured with nATC or T-GFPor T-CD3εζ28.-GFP at different E:T ratios for 24 h. The viability of cancer cells was determined by using MTS assays. (**D**,**E**) Histograms showed the production of cytokines in different T cells. HeLa-EGFRvIII BiTE cells were co-cultured with nATC or T-GFP or T-CD3εζ28-GFP at different E:T ratios for 24 h, and then the supernatants were collected. The concentration of IL-2 (**D**) and IFN-γ (**E**) in supernatants were detected by enzyme-linked immunosorbent assay (ELISA). (**F**) Histograms showed the expression levels of genes related to T cell activation. HeLa-EGFRvIII BiTE cells were co-cultured with nATC or T-GFPor T-CD3εζ28-GFP at different E:T ratios for 24 h, and then the T cells were collected. The relative expression of CD69, GranzymeB, CD107α, IFNγ and TNFα and housekeeping gene GAPDH was evaluated by RT-qPCR assay. The fold change of each gene was calculated using the formula. (**G**,**H**) Histograms showed the expansion of T cell during sequential stimulated by cancer cells. The different T cells were repeatedly stimulated with HeLa-EGFRvIII BiTE cells at an E:T ratio of 5:1 for 24 h. Then T cells were harvested and cultured in fresh medium for 7 days. On the day8, the cultured different T cells were repeatedly stimulated with HeLa-EGFRvIII BiTE cells at an E:T ratio of 5:1 for 24 h. Then T cells were harvested and cultured in fresh medium for another 7 days. The cell numbers of T cells were counted on day 1, 3, 5 and 7 during first 7-day round (**G**) and second 7 day round (**H**), respectively. Statistical differences were assessed by mutual comparison between each two groups using multiple comparisons *t* test. (* *p* < 0.05, ** *p* < 0.01, *** *p* < 0.001, ns: no significance).

**Figure 3 cancers-14-04947-f003:**
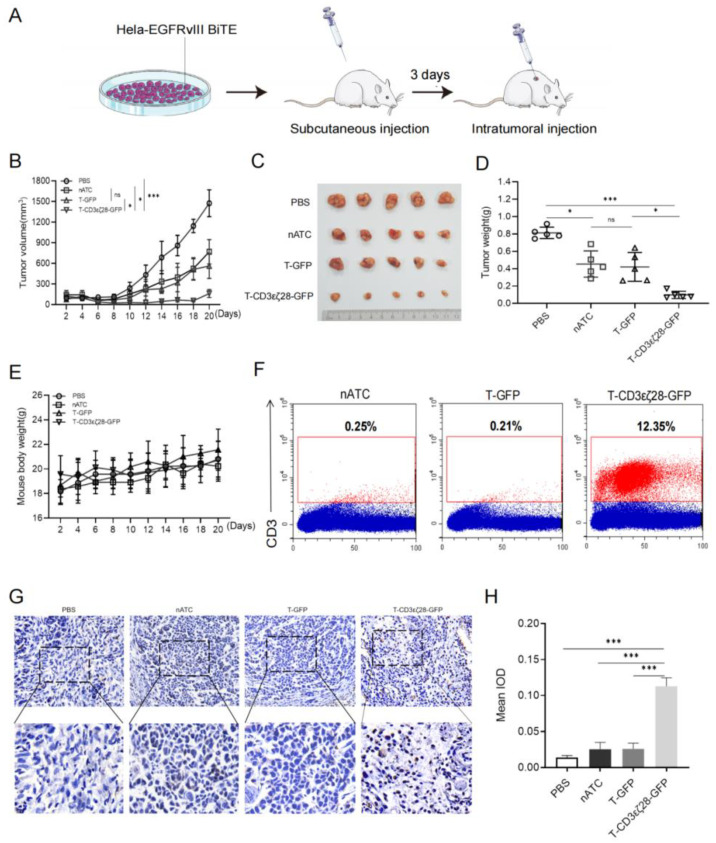
T-CD3εζ28-GFP had enhanced antitumor activity in the xenograft tumor model. (**A**) Schematic diagram of an HeLa-EGFR vIII BiTE tumor model treated with T cells. Nude mice were subcutaneously injected HeLa-EGFR vIII BiTE cells (5 × 10^6^ tumor cells per animal in 100 μL PBS) on day 0. Control 1 × PBS and nATC, T-GFP, T-CD3εζ28-GFP cells (1 × 10^7^ cells per animal in 100 μL PBS) were intra-tumorally injected into mice on day 3. (**B**)The tumor growth curves of different group are displayed. After T cells was injected, the tumor sizes were measured with calipers every two days, and tumor volumes (in mm^3^) were determined using the formula W^2^ * L/2, where W and L represents tumor width and tumor length, respectively. (**C**,**D**) Comparison of tumor sizes and weights in different treated groups at the end of the experiment. All mice were sacrificed on the day 21. The sizes of all tumors were orderly exhibited (**C**). The tumor weights were presented and compared in different groups (**D**). (**E**) Comparison of the mice weights in different treated group. After T cells was injected, the mice weights were weighed every two days. (**F**) Flow cytometry and immunohistochemistry analyses show the percentages and densities of human T cells in tumor tissues, respectively. On day 9 after T cell injection, one mouse from each indicated group was sacrificed and took out the tumor tissues. The single cell suspensions were prepared from tumor tissues, and stained with anti-hCD3 mAb for FCM. (**G**,**H**) The tumor tissues embedded in paraffin wax were stained with anti-hCD3 mAb and observed under microscope. The distributions of T cells (CD3 positive) were displayed in the pictures (**G**). The CD3 positive cell density were evaluated according to the image gray-scale analysis (**H**). In this case, mAb, monoclonal antibody; IOD, integral optical density. Statistical differences were determined by mutual comparison between each two groups using multiple comparisons *t* test. (* *p* < 0.05, *** *p* < 0.001, ns: no significance).

**Figure 4 cancers-14-04947-f004:**
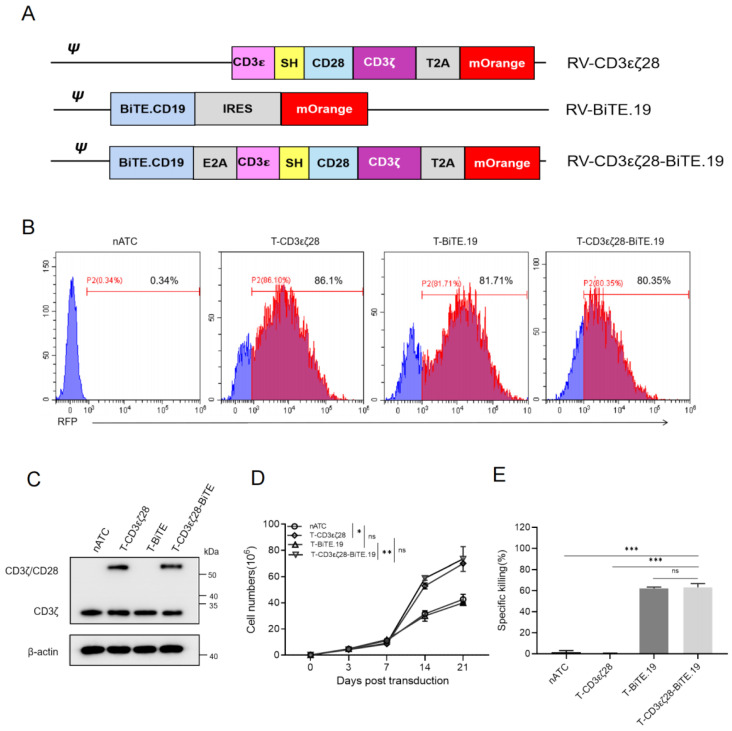
Generation of T-CD3εζ28-BiTE.19 cells. (**A**) Schematic diagram representing different engineering T cells. (**B**) Flow cytometry analysis showed the transduced rate of retrovirus carrying CD3εζ28-mO or BiTE-mO or CD3εζ28-BiTE-mO in T cells. nATC was a non-transduced control. (**C**) The expression levels of endogenous CD3ζ and CD3εζ28 fusion protein in different T cells were evaluated by western blotting with anti-CD3ζmAb. (**D**) The growth curves of different T cells were made based on the cell number counting at indicated timepoint for 21 days in vitro. (**E**) Histograms display the percentages of specific cancer cell killings of nATC cells mediated by BiTE targeting CD19. The supernatant from cultured nATC, T-CD3εζ28, T-BiTE.19 and T-CD3εζ28-BiTE.19. cells were collected. Then the nATC cells were co-cultured with Hela cells expressing CD19 (Hela-CD19) in a 2:1 ratio supplemented with the collected cell supernatant for 24 h. The viability of co-cultured Hela-CD19 cells was determined by MTS assay. Statistical differences were assessed by mutual comparison between each two groups using multiple comparisons *t* test. (* *p* < 0.05, ** *p* < 0.01, *** *p* < 0.001, ns: no significance).

**Figure 5 cancers-14-04947-f005:**
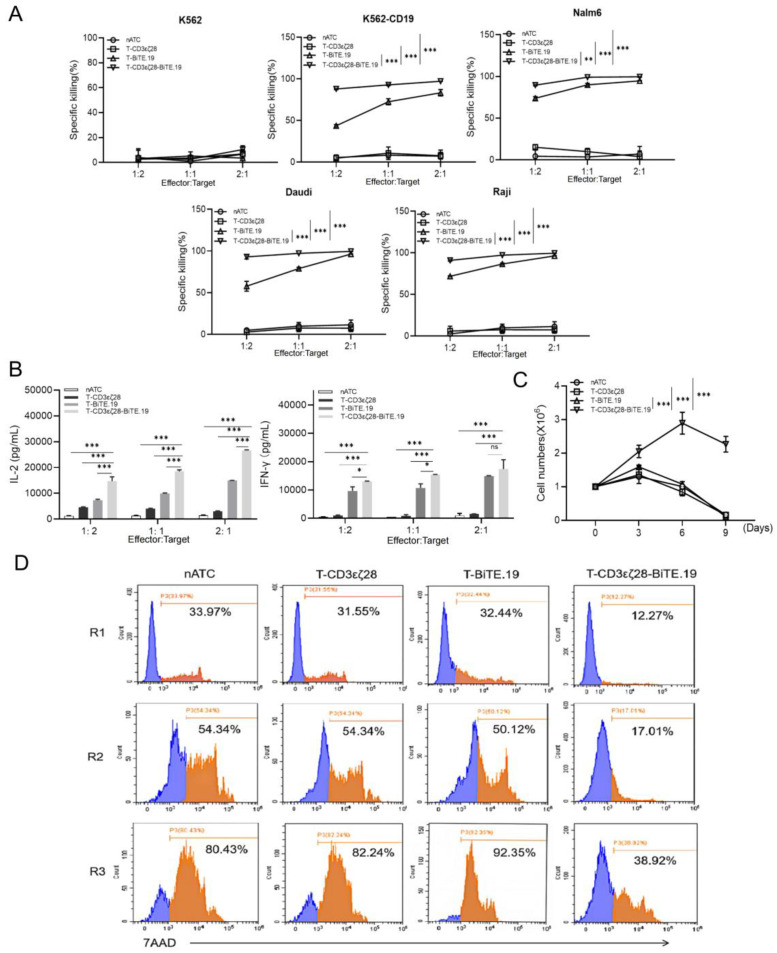
Evaluation of the cancer cell killing effects and persistence of T-CD3εζ28-BiTE.19 in vitro. (**A**) Comparison of the cancer cell killing effects of different T cells. Different cancer cells were co-cultured with nATC or T-CD3εζ28 or T-BiTE.19 or T-CD3εζ28-BiTE.19 at indicated effector to target (E:T) ratios 24 h. The viability of cancer cells was determined by luciferase assays. (**B**) Histograms showed the production of cytokines in different T cells. Nalm6 cells were co-cultured with nATC or T-CD3εζ28 or T-BiTE.19 or T-CD3εζ28-BiTE.19 at different E:T ratios for 24 h, and then the supernatants were collected. The concentration of IL-2 and IFN-γ in supernatants were detected by enzyme-linked immunosorbent assay (ELISA). (**C**,**D**) Histograms showed the expansion of T cell during sequential stimulated by cancer cells. The T cells were repeatedly stimulated with Hela-CD19 at an E:T ratio of 5:1 every 3 days. The cell numbers and apoptosis of T cells were counted (**C**) and evaluated by FCM (**D**) on day 3, 6 and 9, respectively. Statistical differences were assessed by mutual comparison between each two groups using multiple comparisons *t*-test. (* *p* < 0.05, ** *p* < 0.01, *** *p* < 0.001, ns: no significance).

**Figure 6 cancers-14-04947-f006:**
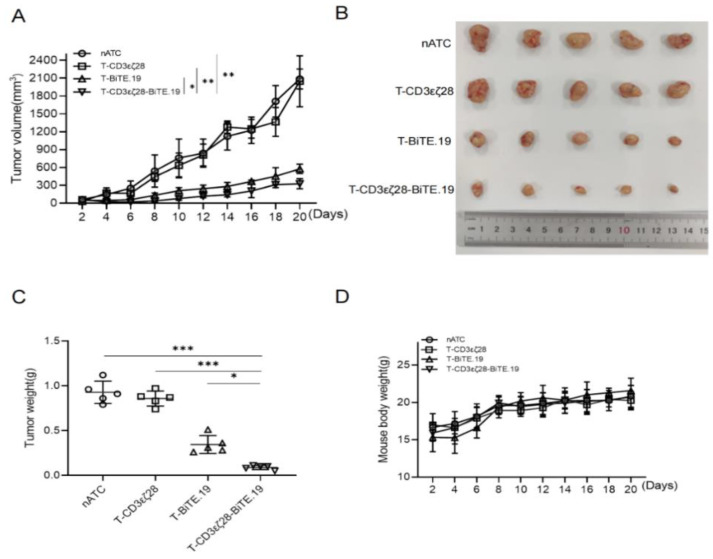
Evaluation of the anti-tumor effects of T-CD3εζ28-BiTE in the xenograft tumor model. The subcutaneous xenograft tumor models were established by injecting Hela-CD19 cells, 5 × 10^6^ cells per animal in 100 μL PBS. After 3 days, cells, 1 × 10^7^ T cells per animal were intra-tumorally injected. (**A**) The tumor growth curves of each group are displayed. At the end of the experiment, the tumors tissues were dissected and analyzed by measuring tumor size (**B**) and weight (**C**), and mice weights were measured (**D**) in grams. (* *p* < 0.05, ** *p* < 0.01, *** *p* < 0.001).

## Data Availability

The data presented in this study are available in this article (and Appendix A).

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
