# Peer review of "Combination of a Novel Fusion Protein CD3εζ28 and Bispecific T Cell Engager Enhances the Persistance and Anti-Cancer Effects of T Cells"

_cancers, 2022, doi:10.3390/cancers14194947_

Round 1
Reviewer 1 Report
Yu et al., generated effector tumor-killing T cells combining the fusion protein CD3εζ28 and BiTE. The new T cells demonstrated efficient antitumor efficacy and showed promise for cancer therapy. However, some questions need to be addressed.
1. Figure 1B shows about 60-70% GFP expression on engineered T cells. Were these T cells pre-sorted for GFP+and then used for the following killing assays?
2. For Figure 2A,C,D and E, 5A and 5B. Please indicate the X-axis label in the figure.
3. Please check through the figures and correct all labeling, for example, IL-2 not IL2, IFN-γ not IFNγ, Granzyme B not GranzymeB, TNF-α not TNFα, pg/mL not pg/ml, *p < 0.05 not *p<0.05. In the figure legends, the authors used both (E) and (e) to cite the figures. Typos and small mistakes should be avoided.
4. For Figure 2, a BiTE-negative Hela cell line should be included and repeated for the killing/functional assay, to confirm the killing specificity of T-CD3εζ28-GFP cells.
5. Figure 3 and 6, please describe the rationale of preforming intratumoral injection instead of i.v. injection of T cells.
6. Figure 3F, the tumor-infiltrating T cells should be compared with pre-infusing T cells, and functional and effector markers should be compared, such as PD-1, CTLA-4, TIM-3, CD69, CD25, CD62L, etc.
7. Figure 5D, T cell exhaustion markers should be stained and compared.
8. Figure 6, the tumor-infiltrating T cells should be detected using flow cytometry.
9. Figure 6, a relevant tumor model (Raji or HL60) should be used for the in vivo experimental. Hela-CD19 cell line is not clinically relevant. The authors should discuss and explain the rationale.
Author Response
- Figure 1B shows about 60-70% GFP expression on engineered T cells. Were these T cells pre-sorted for GFP+ and then used for the following killing assays?
Answers: The GFP+ T cell were not sorted for the following killing assays. Indeed, CAR-T cells are not sorted in most CAR-T cell therapy studies.
- For Figure 2A,C,D and E, 5A and 5B. Please indicate the X-axis label in the figure.
Answers: We have added the X-axis labels of Figure 2A,C,D and E, 5A and 5B.
- Please check through the figures and correct all labeling, for example, IL-2 not IL2, IFN-γ not IFNγ, Granzyme B not GranzymeB, TNF-α not TNFα, pg/mL not pg/ml, *p < 0.05 not *p<0.05. In the figure legends, the authors used both (E) and (e) to cite the figures. Typos and small mistakes should be avoided.
Answers: We have changed all the labellings.
- For Figure 2, a BiTE-negative Hela cell line should be included and repeated for the killing/functional assay, to confirm the killing specificity of T-CD3εζ28-GFP cells.
Answers: Figure 2A is the experiments data based on the killing assay of the BiTE-negative Hela cell. We cocultured the T cells and the BiTE-negative Hela cells with or without the BiTE to confirm the killing specificity of T-CD3εζ28-GFP cells.
- Figure 3 and 6, please describe the rationale of preforming intratumoral injection instead of i.v. injection of T cells.
Answers: we chose intratumoral injection to increase the initial tissue concentration of the drugs. Because intratumoral drugs may be easy to enter the tumor-draining lymph nodes, and further enhancing the local antitumor immune response. We will compare the therapeutic effects of intratumoral injection with i.v. injection in future studies.
- Figure 3F, the tumor-infiltrating T cells should be compared with pre-infusing T cells, and functional and effector markers should be compared, such as PD-1, CTLA-4, TIM-3, CD69, CD25, CD62L, etc.
Answers: our aim for Figure 3 is to determine whether combination of CD3εζ28 and BiTE can enhance the survive, persistence and antitumor effects of T cells in vivo. So our experiments are designed to focus on these aspects. In addition, these animal experiments have been finished long time before, and it is very difficult for us to order so many animals to repeat these experiments due to the serious covid-19 situation.
- Figure 5D, T cell exhaustion markers should be stained and compared.
Answers: our aim for Figure 5 is to determine whether T cells both expressing CD3εζ28 and BiTE have enhanced persistence and tumor cell killing effects in vitro. So our experiments are designed to focus on detecting the index to reflect the survive and tumor cell killing effects of T cells.
- Figure 6, the tumor-infiltrating T cells should be detected using flow cytometry.
Answers: our aim for Figure 6 is to determine whether T cells both expressing CD3εζ28 and BiTE have strong antitumor effects in vivo. So our experiments are designed to focus on these aspects. In addition, these animal experiments have been finished long time before, and it is very difficult for us to order so many animals to repeat these experiments due to the serious covid-19 situation.
- Figure 6, a relevant tumor model (Raji or HL60) should be used for the in vivo experimental. Hela-CD19 cell line is not clinically relevant. The authors should discuss and explain the rationale.
Answers: At present, the immunotherapy against lymphoma has achieved very good results. However, in solid tumors, immunotherapy is not very effective. We need to develop more therapies targeted to solid tumors. Therefore, we chose Hela cells to construct a solid tumor model.
Reviewer 2 Report
Bispecific T cell engager (BiTE) has achieved great success in the tumor treatment. The first-in-class BiTE, blinatumomab, has been approved for the treatment of patients with relapsed and/or refractory B cell-precursor acute lymphoblastic leukemia. In the manuscript, the authors generated a chimeric fusion protein CD3εζ28 and tried to enhance the antitumor effect of BiTE via activating T cells. Although this article seems like a good one, there are some concerns should be fully addressed.
1. The authors claimed that combination of CD3εζ28 and BiTE could be an effective approach to improve the effector function of T cells mediated by BiTE. However, T cells genetically modified to express the chimeric fusion protein CD3εζ28 will strictly limit the clinical application of this strategy. The authors should point out and discuss this limitation.
2. In the xenograft tumor model, T cells were intratumorally injected to verify the anti-tumor effect of T-CD3εζ28-BiTE.19. The authors reported that T-CD3εζ28-BiTE.19 cells could effectively inhibit tumor growth. However, intratumorally injection isn’t the general method for adoptive cell therapy. The antitumor effect of T-CD3εζ28-BiTE cells should be confirmed by intravenous injection.
3. Figure legend of Figure 3H missed or there is an error in the Figure legend.
Author Response
- The authors claimed that combination of CD3εζ28 and BiTE could be an effective approach to improve the effector function of T cells mediated by BiTE. However, T cells genetically modified to express the chimeric fusion protein CD3εζ28 will strictly limit the clinical application of this strategy. The authors should point out and discuss this limitation.
Answers: At present, cancer therapy through genetically modified T cells is well established. CAR-T and TCR-T are all based on this way. Therefore, genetically modifying T cells by expressing fusion proteins is not a difficult task.
- In the xenograft tumor model, T cells were intratumorally injected to verify the anti-tumor effect of T-CD3εζ28-BiTE.19. The authors reported that T-CD3εζ28-BiTE.19 cells could effectively inhibit tumor growth. However, intratumorally injection isn’t the general method for adoptive cell therapy. The antitumor effect of T-CD3εζ28-BiTE cells should be confirmed by intravenous injection.
Answers: we chose intratumoral injection to increase the initial tissue concentration of the drugs. Because intratumoral drugs may be easy to enter the tumor-draining lymph nodes, and further enhancing the local antitumor immune response. We will compare the therapeutic effects of intratumoral injection with i.v.injection in future studies.
- Figure legend of Figure 3H missed or there is an error in the Figure legend.
Answers: We have corrected the legend of Figure. 3H.
Round 2
Reviewer 1 Report
The manuscript could be accepted in present form.
Author Response
Thank you very much for spending much time to review our manuscript!
Reviewer 2 Report
Intratumorally injection of T cells isn’t the general method for adoptive cell therapy. The in vivo antitumor effect of T-CD3εζ28-BiTE cells should be confirmed by intravenous injection.
Author Response
Thank you very much for your very crucial suggestion. The main aim of our study is to determine whether the fusion protein CD3εζ28 can enhance the anti-tumor effect of T cell mediated by BiTE. Althrough intravenous injection is a common approach to determine the antitumor effect of T cell based therapy in clinic and clinical trails, intratumorally injection of T cells is also a common method to be used for preclinical studies. Moreover, some practical issues, e.g funds and covid-19 situation, make it very difficult for us to repeat the animal experiments using intravenous injection. However, we plan to perform the in vivo test using intravenous injection in future studies.
Round 3
Reviewer 2 Report
None